# Delphi studies in social and health sciences—Recommendations for an interdisciplinary standardized reporting (DELPHISTAR). Results of a Delphi study

**Marlen Niederberger**[1]*, Julia Schifano[1], Stefanie Deckert[2], Julian Hirt[3,4], Angelika Homberg[5], Stefan Köberich[6], Rainer Kuhn[7,8], Alexander Rommel[9], Marco Sonnberger[10], the DEWISS network[¶]

1 Department of Research Methods in Health Promotion and Prevention, Institute for Health Sciences, University of Education Schwäbisch Gmünd, Schwäbisch Gmünd, Germany, 2 Center for Evidence-Based Healthcare, University Hospital and Medical Faculty Carl Gustav Carus, TU Dresden, Dresden, Germany, 3 Department of Health, Institute of Nursing Science, Eastern Switzerland University of Applied Sciences, St. Gallen, Switzerland, 4 Pragmatic Evidence Lab, Research Center for Clinical Neuroimmunology and Neuroscience Basel (RC2NB), University Hospital Basel and University of Basel, Basel, Switzerland, 5 Department of Medical Education Research, Medical Faculty Mannheim at Heidelberg University, Mannheim, Germany, 6 Medical Center, Faculty of Medicine, University of Freiburg, Freiburg, Germany, 7 DIALOGIK Non-Profit Institute for Communication and Cooperation Research, Stuttgart, Germany, 8 Center for Interdisciplinary Risk and Innovation Studies (ZIRIUS), University of Stuttgart, Stuttgart, Germany, 9 Department 2, Epidemiology and Health Monitoring, Robert Koch-Institut, Berlin, Germany, 10 Department of Sociology of Technology, Risk and Environment, University of Stuttgart, Stuttgart, Germany

¶ The members of the DEWISS network can be viewed at the following website: https://delphi.ph-gmuend.de/.

* marlen.niederberger@ph-gmuend.de

## Abstract

### Background

While different proposals exist for a guideline on reporting Delphi studies, none of them has yet established itself in the health and social sciences and across the range of Delphi variants. This seems critical because empirical studies demonstrate a diversity of modifications in the conduct of Delphi studies and sometimes even errors in the reporting. The aim of the present study is to close this gap and formulate a general reporting guideline.

### Method

In an international Delphi procedure, Delphi experts were surveyed online in three rounds to find consensus on a reporting guideline for Delphi studies in the health and social sciences. The respondents were selected via publications of Delphi studies. The preliminary reporting guideline, containing 65 items on five topics and presented for evaluation, had been developed based on a systematic review of the practice of Delphi studies and a systematic review of existing reporting guidelines for Delphi studies. Starting in the second Delphi round, the experts received feedback in the form of mean values, measures of dispersion, a summary of the open-ended responses and their own response in the previous round. The final draft

**Data Availability Statement:** All relevant data are within the manuscript and its Supporting Information files.

**Funding:** The authors are members of the DEWISS network. The DEWISS Network is supported by the German Research Foundation (project number: 429572724). The funders had no role in the study design, data collection and analysis, decision to publish, or preparation of the manuscript.

**Competing interests:** The authors have declared that no competing interests exist.

of the reporting guideline contains the items on which at least 75% of the respondents agreed by assigning scale points 6 and 7 on a 7-point Likert scale.

## Results

1,072 experts were invited to participate. A total of 91 experts completed the first Delphi round, 69 experts the second round, and 56 experts the third round. Of the 65 items in the first draft of the reporting guideline, consensus was ultimately reached for 38 items addressing the five topics: Title and Abstract (n = 3), Context (n = 7), Method (n = 20), Results (n = 4) and Discussion (n = 4). Items focusing on theoretical research and on dissemination were either rejected or remained subjects of dissent.

## Discussion

We assume a high level of acceptance and interdisciplinary suitability regarding the reporting guideline presented here and referred to as the "Delphi studies in social and health sciences–recommendations for an interdisciplinary standardized reporting" (DELPHISTAR). Use of this reporting guideline can substantially improve the ability to compare and evaluate Delphi studies.

## Introduction

Internationally, Delphi studies have proven themselves in a variety of disciplines and fields of application. Analyses show a growing prevalence of this technique, especially in the contexts of medicine, science and technology, and the social sciences [1]. They represent an important tool for analyzing potential future conditions [2, 3]. Associated with this is the idea of collective intelligence, according to which the prognostic ability of a group of experts is better than that of a single expert [4]. In the context of health sciences research, Delphi studies are used in the medical and natural sciences [5] and the behavioral social sciences [6]. They are selected for use if little or inconsistent evidence is available [7], or primary studies are not possible because of economic, ethical, or pragmatic reasons, or there are practical challenges in clinical or nursing contexts.

Due to the prevalence of Delphi studies [1, 8], different authors have already formulated proposals for reporting Delphi studies [9–12]. One guideline has been published using the acronym CREDES (Guidance on Conducting and REporting DElphi Studies) [9]. Another has been published using the keyword ACCORD (ACcurate COnsensus Reporting Document) [13, 14]. Yet none of these reporting guidelines claims to be valid for the many diverse areas of application or Delphi variants in the health and social sciences. This gap should be closed with the help of the study presented here, in that we develop the reporting guideline "DELPHISTAR—Delphi studies in social and health sciences—recommendations for an interdisciplinary standardized reporting."

### Characteristics and variants of Delphi techniques

Delphi techniques are structured survey procedures in which complex topics, on which uncertain or incomplete knowledge exists, are evaluated by experts in an iterative process [15]. Specific to a Delphi procedure is that the survey is repeated and, from the second survey round onwards, information is shared regarding the results of the previous round enabling the

respondents to reconsider their judgments and, if needed, revise them. Five typical characteristics of the Delphi process can be gleaned from the methods literature [7, 16]:

1. Experts are surveyed while typically preserving their anonymity.

2. The survey is conducted in at least two Delphi rounds.

3. A standardized questionnaire is used, often with open-ended questions to gather arguments and capture the horizons of legitimation.

4. The statistical analysis is based on descriptive calculations.

5. From the second Delphi round onwards, the experts receive feedback on the results of the previous round along with the questionnaire and can thus reconsider and, if necessary, revise their judgments.

Some authors define the Delphi process more narrowly and focus on the finding of consensus among the expert judgments [17, 18]. According to Dalkey and Helmer [19], the process is suitable "to obtain the most reliable consensus of opinion of a group of experts. . .by a series of intensive questionnaires interspersed with controlled feedback." Narrowing the definition to consensus, however, seems discriminating given the many different settings in which Delphi studies are applied, for instance, to forecast future developments [3] or discover and aggregate knowledge [20].

In recent years many variations of the Delphi procedure have been developed [21, 22]. More than 10 different variants have already been identified [23, 24]. The Delphi variants differ from each other in terms of process design, for instance, whether or not the Delphi rounds are held separately or overlap with each other, in the weighting of open-ended and standardized responses, and also in regard to the expert panel, e.g., group size and the handling of anonymity [24, 25]. Among the Delphi variants are both established variants and some that have hardly been used before:

- *Real-time Delphi*, in which expert judgments are reflected back online and in real time. There are no clearly separate Delphi rounds [21, 26].

- *Delphi markets*, where the Delphi concept is combined with virtual marketing platforms (prediction markets) and the findings of Big Data research to improve abilities to forecast the future and the quality on which such predictions are based [27].

- *Policy Delphis* are concerned with capturing dissent, meaning a wide range of diverse judgments [16, 28].

- *Argumentative Delphi*, where the focus is on the qualitative reasoning for the experts' quantitative evaluations [23].

- *Group Delphi*, for which the experts are invited to a workshop to openly formulate and discuss arguments in favor of divergent judgments [29, 30].

- *Deliberative Delphi* (citizens' Delphi), in which citizens are surveyed iteratively. In between the Delphi rounds, they are trained to make informed and responsible judgments [31].

- *Fuzzy Delphi* applies different analytical strategies to quantify the linguistic labels often used in the Likert scales to allow for potential differences in the understanding of these expressions when calculating mean values [32].

- *Café Delphi*, in which a smaller number of experts are surveyed in an informal, "café-like" atmosphere [33].

A look at the paper published by Mullen in 2003 [34] makes it clear that this list here is far from complete. She identifies more than ten additional Delphi variants (e.g., Delphi conference, decision Delphi, Delphi forecast, ranking Delphi), but without defining them more closely or differentiating them from one another. Furthermore, different systematic reviews report on countless other, hardly nameable or understandable, modifications of Delphi procedures [9, 35].

The differentiation between Delphi variants is accompanied by epistemological and methodological specifications regarding the classic Delphi design, which also affects the characteristics. Hence, the definition of "expert" is broadened to include not only people in certain professional positions or who have attained academic excellence, but also people with a specific kind of lifeworld experience, which then means that experts are not just members of certain professions, but also patients, patients' relatives, or users [36, 37].

From an epistemological standpoint, newer Delphi studies are often based on constructivist assumptions and use not only standardized questionnaires, but also explorative instruments in the form of open central questions [38] or workshops [39]. Ensuring anonymity, however, remains a constant in the evolution of the Delphi technique; the names of participating experts are published only in exceptional cases [30, 40].

Given the often considerably limited scope of journal articles, it is sometimes impossible to present and justify the use of the selected Delphi variant and any modifications to it, such that it is all sufficiently transparent to outsiders. In the following, a look at publication practices suggests, at the least, how these aspects are addressed.

## Reporting Delphi studies

Different systematic reviews document unclear or potentially misleadingly formulated approaches in Delphi studies [41]. There are sometimes even errors in the presentation of the method or statistical analysis [42]. As an example, even a survey of experts in a single round is declared to be a Delphi study [43]. In respect to presenting the methodological approach, questions remain unanswered, for instance, regarding the form of feedback [44], why the selected number of rounds was chosen [45], at what point "consensus" was defined [46], and how high the response rate for each Delphi round was [47].

A recognized reporting guideline can help to counteract such methodological misunderstandings and imprecisions. Ultimately, the quality of Delphi studies can also be improved through more transparency. This is the aim pursued by the present study concerning the development of the reporting guideline "DELPHISTAR—Delphi studies in social and health sciences—recommendations for an interdisciplinary standardized reporting."

## Background

The scientific network DEWISS has set the goal of developing a reporting guideline for Delphi studies that is valid for the different Delphi variants and diverse fields within the health and social sciences (more information is available at https://delphi.ph-gmuend.de/). The German-speaking DEWISS Network is comprised of 20 scientists and academics from different subject areas and disciplines. All of the members conduct Delphi studies in the context of their research and grapple with the methodological and epistemological aspects of Delphi techniques. They perform methodological tests, carry out surveys to improve the methodological basis of Delphi studies, advise other researchers on how to conduct Delphi studies, and develop concepts and materials that can be used to teach about Delphi procedures (e.g., short videos at https://delphi.ph-gmuend.de/). Since its founding, this network has received funding from the German Research Foundation (Deutsche Forschungsgemeinschaft, DFG), an

overarching institution providing support for science and research in the Federal Republic of Germany (project number 429572724, time period 2020 to 2024).

## Method

Using the acronym DELPHISTAR (OSF registration: https://osf.io/gc4jk), a multi-method research design consisting of three sub-studies was carried out (Fig 1).

- **First sub-study:** In the first step, an overview of Delphi studies was created from a methodological standpoint [41]. A total of 16 previous reviews of Delphi studies were identified, systematically evaluated, and the results summarized in a map [41]. It was seen here that, among other things, there is a diversity of approaches and, in some instances, unexamined modifications to Delphi studies. The research team's awareness of the relevant aspects and the necessity for a reporting guideline was raised by these findings.

- **Second sub-study:** In a systematic review, ten earlier recommendations for reporting Delphi studies were identified, analyzed in terms of content, and examined for commonalities and differences [48]. In the course of this, it was seen, among other things, that these previous recommendations did not claim to have validity across disciplines or for different Delphi variants. The recommendations were often developed for a specific research area, e.g., palliative medicine [9] or medical education [49]. This is possibly the reason why the proposal published in the EQUATOR Network by Jünger et al. [9] did not result in any fundamental improvement in reporting practices [35].

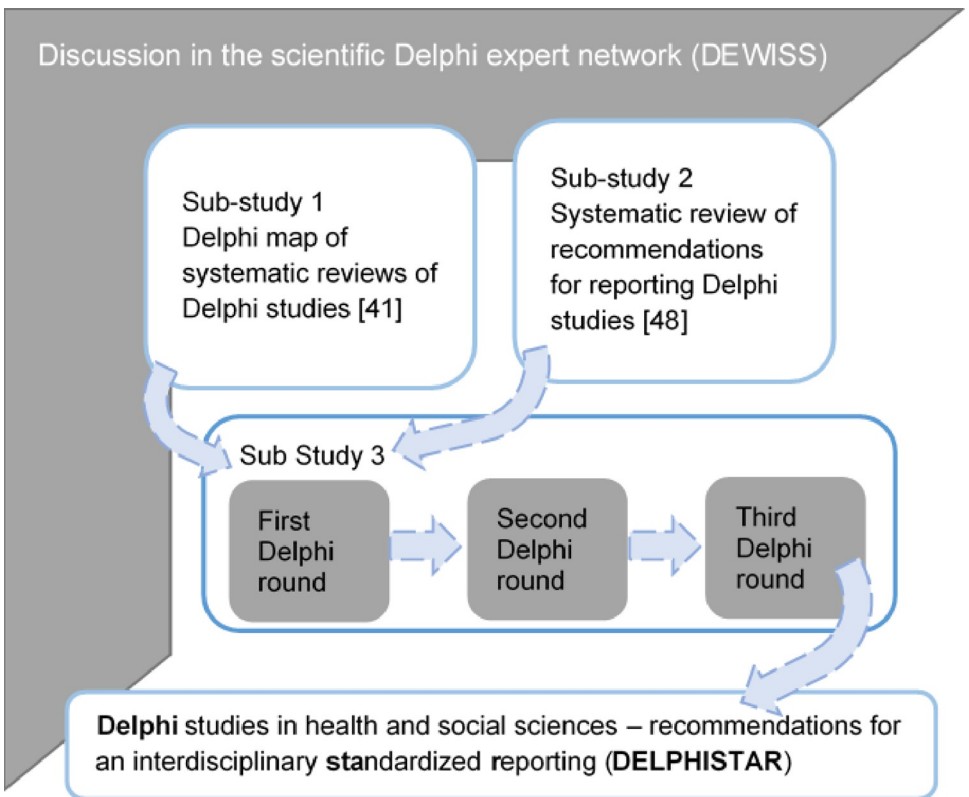

**Fig 1. Methods concept for developing the reporting guideline for Delphi studies.**

- **Third sub-study:** The results gathered from the first two sub-studies were discussed in the DEWISS Network and transformed into a comprehensive reporting guideline for Delphi studies. Consensus among additional Delphi experts was reached on this reporting guideline by means of a Delphi procedure. The selection of the Delphi method is justified by the fact that it is also recommended by other authors for the development of a reporting guideline [50]. The Delphi process is presented in the following.

## The Delphi process

International experts on Delphi procedures were surveyed for the purpose of developing a reporting guideline for Delphi studies. The aim was to find consensus on the reporting criteria. The approach was based on the "classic" Delphi technique with three rounds that were carried out online (Fig 2). Digital collection of data is now an established part of Delphi procedures [25]. However, since our process exhibits the five typical characteristics of a Delphi procedure (see Introduction), we identify our study as a "classic" Delphi. In doing so, we allot a relatively high importance to the free-text responses, in that we analyze them systematically, combine them with the quantitative data, and use them to fine-tune the wording in the reporting guideline.

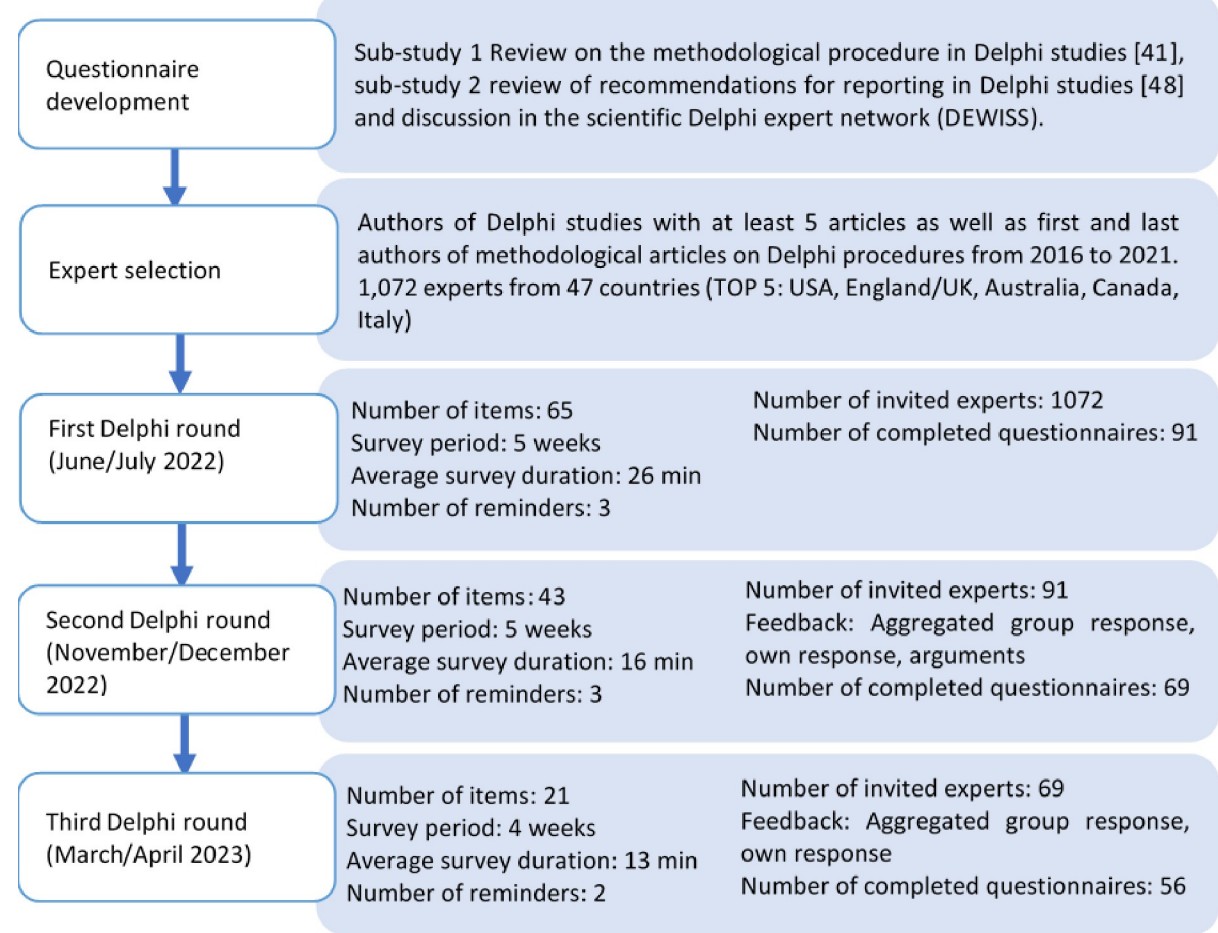

**Fig 2. Process of the Delphi study.**

## Questionnaire development

The questionnaire was developed by the DEWISS Network on the basis of the first two sub-studies [41, 48]. These sub-studies identified existing reporting guidelines and research methods, and the findings were synthesized during several DEWISS network meetings (Table 1). The results were incorporated in the first draft of the reporting guideline for Delphi studies. For this, we selected a structured sequence organized by topics and sections because this resembles established reporting guidelines, particularly the PRISMA guideline for systematic reviews [51].

Finally, items covering five specific topics, each with up to seven sections, were contained in the initial questionnaire (Table 1). They are presented here as they appear in the final version of the reporting guideline.

The proposed content of the reporting guideline was queried in the form of standardized items on a 7-point rating scale ("1 = very unimportant, 7 = very important" or "1 = very unlikely, 7 = very likely") (Fig 3). Different rating scale widths have been established in Delphi studies [9, 52]. Firstly, they enable a separate evaluation of each item; secondly, an experimental study shows that for those taking the survey, the completion time is quicker and the cognitive effort is lower when compared to ranking scales [53]. This is an important argument in regard to participant motivation. With this in mind, we deliberately chose an odd-numbered scale width. Taze et al. [54], to cite one example, also recommend this for Delphi studies. The items were deliberately formulated so that it was possible to understand them without further explanation. Even so, examples were still included in some instances. Each item was programmed as a required question. For this reason, there was always an evasive option available ("cannot evaluate this item").

Also, in all three of the rounds the experts were asked in a standardized manner about the certainty of their judgment ("1 = extremely uncertain, 7 = absolutely certain") so that this could be taken into consideration in the analysis. In the first and second Delphi rounds it was possible to comment freely after each topic (see Fig 3). The free-text boxes were each limited to 300 characters. In the third and final survey round it was possible to comment freely at the end of the survey without any limitations on the character count.

**Table 1. Reporting guideline.** Overview of the items that were evaluated according to topic and section.

| Topic | Section (n = Anzahl der Items) |
|---|---|
| Title and abstract (n = 3) | Title and abstract (n = 3) |
| Context (n = 16) | Formal (n = 8) |
| | Theory (n = 3) |
| | Content (n = 5) |
| Method (n = 32) | Body of knowledge & Integration of knowledge (n = 3) |
| | Delphi variations (n = 2) |
| | Sample of experts (n = 5) |
| | Survey (n = 11) |
| | Delphi rounds (n = 3) |
| | Feedback (n = 4) |
| | Data analysis (n = 4) |
| Results (n = 4) | Delphi process (n = 5) |
| | Results (n = 1) |
| Discussion and dissemination (n = 8) | Quality of findings (n = 5) |
| | Dissemination (n = 3) |

## Data analysis and results

**V15:** If Delphi studies and their results are reported, how important do you consider the following aspects of the "Data analysis and results - Section: Results"?

Please respond with a "1" if you view an aspect as very unimportant or with a "7" if you view it as being very important. You can use the numbers in between to graduate the scale. You may also indicate that you cannot or do not wish to evaluate a particular item.

| | 1 very unimportant | 2 | 3 | 4 | 5 | 6 | 7 very important | cannot evaluate this item |
|---|---|---|---|---|---|---|---|---|
| 15.a Presentation of the results for each Delphi round and the final results | ○ | ○ | ○ | ○ | ○ | ○ | ○ | ○ |
| 15.b Information about how dissent and unclear results were handled | ○ | ○ | ○ | ○ | ○ | ○ | ○ | ○ |

**S13-15:** How certain are you in responding to the topic "Data analysis and results" of the Reporting Guideline?

Please respond with a "1" if you are extremely uncertain or with a "7" if you are absolutely certain. You can use the numbers in between to graduate the scale.

| 1 extremely uncertain | 2 | 3 | 4 | 5 | 6 | 7 absolutely certain |
|---|---|---|---|---|---|---|
| ○ | ○ | ○ | ○ | ○ | ○ | ○ |

**O13-15:** In the following you have the option to give reasons for your responses and to provide additional information.

Please use the text box below. Please note: The text box is limited to 300 characters!

Note: For orientation purposes, the research team structured the questions according to an internal numbering system for the items (V15, S13-15, O13-15).

**Fig 3. An example of a page from the questionnaire on judgment certainty and a text box for comments (Source: Unipark).**

Also integrated into the survey were questions about the respondents' expertise (discipline, country, experience with Delphi studies, proficiency as a Delphi practitioner). These served to describe the sample.

The survey was conducted in English. The initial questionnaire, including the reporting guideline, was translated by a native English speaker and then reviewed for accuracy by methods experts at the Leibniz Institute for the Social Sciences (GESIS), a renowned German research institute in the empirical social sciences. In all three of the Delphi rounds experts were requested not to use any machine translation tools in order to avoid any distortions as a result of translation errors.

The comprehensibility of the questions and the technical functioning of the online survey were tested prior to each Delphi round by DEWISS Network members who had not directly collaborated in the questionnaire development.

## Selecting the experts

Considered as experts were academics who had conducted several Delphi studies themselves and/or who were working on methodological issues related to the Delphi technique. These experts were identified via publications. A search was conducted of two databases compiled by the DEWISS Network and freely accessible through ZOTERO [48]. The first database contains Delphi primary studies (available at: https://www.zotero.org/groups/4396781/dewiss_ datenbanken_delphi-studien/collections/25H44TFI), and the second has publications based on the methodology of Delphi studies (e.g., reviews, methods experiments; available at: https:// www.zotero.org/groups/4396781/dewiss_datenbanken_delphi-studien/collections/ NGTBI3PE). Both databases were created in 2021 based on systematic research of the literature in the central databases for health and social sciences (Scopus, MEDLINE via PubMed, CINAHL and Epistemonikos) and contain Delphi studies and methods papers published between 2016 and 2021. The search was conducted using the keyword "delphi*" in the title or abstract. Publications were included if they involved methodological publications regarding Delphi studies or Delphi primary studies in the health or social sciences. The collection of methods-based studies includes 155 papers and the one with Delphi primary studies comprises 7,044 papers [48]. Authors who had published at least five papers (n = 863) were filtered out of the primary study collection. All lead and senior authors (n = 228) were filtered out of the database containing the methods studies. Nineteen authors were present in both databases so that, in the end, 1,072 Delphi experts were identified and invited to participate in the Delphi study. The author information listed in the publications was used as the contact information. The sample contained 352 women and 710 men (10 unclear) from 47 countries (TOP 5: USA, England/UK, Australia, Canada, Italy).

Participation in the Delphi study was voluntary and anonymous. Informed consent was obtained from all of the participants at the beginning of the survey using an online form. The study design complies with the Helsinki Declaration [55], with regard for the European General Data Protection Regulation [56] and the principles of the DFG [57].

## Data collection

The programming and sending of the questionnaire was done using Unipark software [58]. The invitation email contained a personalized link to the questionnaire and a PDF attachment with the contents of the reporting guideline that were to be evaluated. The time period for the survey was always a minimum of four weeks, during which two to three reminders to participate in the Delphi study were sent (Fig 2). Along with each survey questionnaire, the experts also received a PDF of the preliminary reporting guideline. Each time it was made clear which items had been agreed on, which items had been reworded, and if any new items had been added.

**First Delphi round.** All of the identified experts (n = 1,072) were invited by email to participate in the first Delphi round. Due to security rules at some institutions, some of the emails were blocked, which is why only 87% (n = 934/1,072) of the emails were deliverable.

**Second Delphi round.** The initial questionnaire was revised based on the results of the first Delphi round, meaning that consented items were removed and the remaining items were reworded as necessary based on the free-text comments. The changes in wording were highlighted in color so that the experts could see and understand them. The revisions served to fine-tune the semantics and validate the changes by passing them back to the surveyed experts [59]. This approach is often described in "classic" Delphi studies [60, 61].

The experts received feedback on the statistical group response (aggregated percent agreement on the scale points 6+7, mean value, standard deviation) from the previous round and a

summary of the arguments made in the open-ended responses. In addition, the experts were able to see their own responses to the standardized items from the previous round. Furthermore, the definition of consensus was also communicated to the experts.

Experts who had completed the first round were contacted one week before the second Delphi round informing them about it and requesting them to participate again.

**Third Delphi round.** The questionnaire was revised anew and shortened based on the results of the second Delphi round. Shortening the questionnaire was also undertaken as a measure to maintain participants' motivation to participate.

As feedback, experts received the statistical group response from the previous round and again were able to see their own responses to the standardized items. Since there were only a few new arguments in the open-ended responses and these had been integrated into the questionnaire as part of the revision process, no summary of the arguments made in the open-ended responses was included with the questionnaire at this point in the process. Changes in the wording were, however, again made visible using color highlighting.

## Data analysis

Statistical analysis was performed using R [62]. The responses to the standardized questions were descriptively analyzed (absolute and relative frequencies, minimum, maximum, mean, median, standard deviation). Consensus was defined *a priori* as follows: Consensus for the inclusion of an item in the reporting guideline exists if at least 75% of the responses assign the scale values of 6 or 7 (very important) on the 7-point rating scale. From the second round onward, all items with a rejection rate of at least 50% were excluded, meaning that less than half of the responses assigned the scale values of 6 or 7 on the 7-point rating scale. Items for which consensus had already been reached were not presented again for evaluation in the subsequent rounds.

Analysis of the open-ended responses from the text boxes was done using the Argument-based QUalitative Analysis strategy (AQUA) [63] with Microsoft Word (2019). The AQUA method is based on established analytical methods in qualitative social research and was developed further for the analysis of qualitative data from Delphi studies. When applying the AQUA method, arguments from the open-ended responses are extracted and categorized by topic [63]. No quantification regarding frequency of mentions was undertaken. The arguments in each Delphi round were discussed in the DEWISS Network and, if needed, used to reword the items on the questionnaire.

## Ethical approval

The ethics commission at the University of Education Schwäbisch Gmünd granted written approval on 10 July 2023, rendering an ethics vote unnecessary.

## Results

Of the 934 experts invited to the first Delphi round, 91 (10%) completed the survey. The second Delphi round had a response rate of 76% (n = 69/91), the third had a response rate of 81% (n = 56/69). Overall, experts hailed from 22 countries (round 1), 20 countries (round 2) and 19 countries (round 3), with about half of the experts working in one of five countries: USA, UK, Canada, Australia and China. The distribution in terms of region and discipline remained comparable for all rounds (Table 2). Between 87% and 89% of the experts in each of the rounds stated that they were associated with the health sciences; the others belonged more to the social sciences (Table 2). The central tendency involving publications by the experts is similar across all of the rounds. The number of Delphi studies personally conducted by the participating

**Table 2. Composition of the expert panel.**

| | | First Delphi round (n = 91) | Second Delphi round (n = 69) | Third Delphi round (n = 56) |
|---|---|---|---|---|
| Land[1] | USA | 18%[2] | 17% | 16% |
| | UK | 11% | 13% | 14% |
| | Canada | 11% | 13% | 13% |
| | Australia | 8% | 10% | 13% |
| | China | 5% | 6% | 7% |
| Disziplin | Humanities | 3% | 3% | 2% |
| | Health science | 87% | 87% | 89% |
| | Engineering science | 2% | 0% | 0% |
| | Other | 8% | 10% | 9% |
| Number of Delphi studies participated in (not as a respondent) | Mean (sd) | 9.5 (9.2) | 17.6 (63) | 18 (67.7) |
| | Median | 6 | 9 | 7.5 |
| Number of Delphi publications | Mean (sd) | 9.6 (15.0) | 11 (18.6) | 10 (15.8) |
| | Median | 6 | 6 | 6 |
| Profiles of expertise on Delphi studies | Delphi beginner | 12% | 10% | 11% |
| | Delphi user | 53% | 51% | 50% |
| | Delphi expert | 35% | 39% | 39% |
| Response behavior | Considered | 34% | 35% | 36% |
| | Intuitive | 12% | 9% | 11% |
| | Sometimes considered/ sometimes intuitive | 51% | 55% | 52% |
| | I can't say | 3% | 1% | 2% |
| Ability of Delphi variants (Scale: 1 = absolutely no ability to 7 = excellent ability)[3] | Scale value 6+7 in %, mean (sd) | | | |
| | Classic Delphi | 68%, 5.9 (1.2) | 72%, 5.9 (1.2) | 75%, 6.0 (1.2) |
| | Real-time Delphi | 20%, 4.3 (1.9) | 20%, 4.2 (2.0) | 23%, 4.4 (1.9) |
| | Group Delphi | 34%, 5.0 (1.8) | 42%, 5.1 (1.9) | 41%, 5.0 (1.8) |
| | Policy Delphi | 19%, 4.0 (2.0) | 17%, 3.9 (2.0) | 21%, 4.1 (2.0) |
| | Argumentative Delphi | 5%, 3.1 (1.8) | 4%, 2.9 (1.8) | 2%, 2.7 (1.6) |
| | Deliberative Delphi | 5%, 3.1 (1.9) | 3%, 2.9 (1.7) | 5%, 2.9 (1.8) |
| | Fuzzy Delphi | 1%, 2.4 (1.4) | 1%, 2.4 (1.5) | 2%, 2.4 (1.5) |

[1]Only the five most frequent countries are listed for this category. [2]The percentages refer to the number of participants in a specific Delphi round. The given values have been rounded, whereby it is possible that rounding differences could result. [3]The question about the ability to apply Delphi variants was not a required question in the first Delphi round.

experts is on average clearly lower in the first Delphi round than in the two subsequent rounds. The results of the self-assessed expert profile and response behavior show only minor fluctuations in the relative frequencies for the rounds (Table 2). The majority of the experts judged their ability to apply classic Delphi techniques as excellent (scale points 6+7 out of 7), whereas less than 50% assessed their abilities to be excellent in regard to the real-time Delphi, group Delphi and policy Delphi. For the other Delphi variants, only 5% or fewer of the experts judged their competence to be high.

All of the judgments were included in the analysis, and the statements on judgment certainty were taken into account when analyzing the items for content and revising the questionnaire because, in all of the Delphi rounds and for all of the topics, the experts on average (median 6) responded with good levels of judgement certainty and the variance among the responses was low (standard deviation ≤1.2).

**Table 3. Results for the topic of *Title and abstract*.**

| No. | Checklist (= Items) | Consensus (Round) | Agreement % (n) |
|-----|---------------------|-------------------|-----------------|
| 1 | Identification as a Delphi procedure in the title | In (R2) | 78% (n = 54) |
| 2 | Identification as a Delphi procedure in the abstract | In (R1) | 96% (n = 87) |
| 3 | Structured abstract (e.g., background, method, results and discussion) | In (R2) | 81% (n = 56) |

*R1/R2/R3 Delphi round 1/2/3; n number; "No." refers to the item number in the final version of the reporting guideline

In total, 65 items were presented for evaluation regarding the reporting guideline. At the end of the three Delphi rounds consensus was found for the inclusion of over 38 items in the reporting guideline for Delphi studies in the health and social sciences (S1 File). The points of agreement and disagreement are discussed below.

### Topic: *Title and abstract*

Consent was reached for all of the items asked about the topic of *Title and Abstract*. The majority of the experts said it is important that Delphi studies can be identified through their titles and abstracts and that the abstract's content should be structured (Table 3).

### Topic: *Context*

The topic of *Context* was covered in three sections: *formal*, *theory* and *content*. For the section on *formal* aspects, it was possible to reach agreement on five items (Table 4). According to the experts' opinions, information about funding sources, author team, methods consulting, project background, and the study protocol are important topics for a Delphi reporting guideline. Dissent exists on whether information about the time point of a Delphi study, an ethics vote,

**Table 4. Results for the topic of *Context*.**

| Section | No. | Checklist (= Items) | Consensus (Round) | Agreement % (n) |
|---------|-----|---------------------|-------------------|-----------------|
| **Formal** | 4 | Information about the sources of funding | In (R2) | 81% (n = 56) |
| | 5 | Information about the team of authors and/or researchers (e.g., discipline, institution) | In (R1) | 76% (n = 69) |
| | 6 | Information about the methods consulting | In (R2) | 75% (n = 60) |
| | 7 | Information about the project's background | In (R2) | 82% (n = 55) |
| | | Time period in which the Delphi study was conducted | *No consensus* | 67% (n = 37) |
| | 8 | Information about the study protocol | In (R1) | 76% (n = 68) |
| | | Information on the ethics vote should be provided. This also includes indicating if no vote was required by the responsible ethics committee | *No consensus* | 67% (n = 33) |
| | | Reference to additional information or materials about the project or Delphi study (e.g., online questionnaire, website on the project background) | *No consensus* | 57% (n = 32) |
| **Theory** | | Positioning within the philosophy of science (e.g., realistic, positivist, constructivist) | Out (R2) | 20% (n = 12) |
| | | Identification of the research paradigm (qualitative or quantitative or Mixed Methods) | *No consensus* | 52% (n = 28) |
| | | Statement of presuppositions (e.g., regarding potentially contradictory topics) | Out (R2) | 46% (n = 29) |
| **Content** | | Highlight why the Delphi study is relevant (e.g., due to research gaps or practical relevance to avoid "research waste") | *No consensus* | 71% (n = 39) |
| | | Reflection on the relevance of the Delphi procedure as a topic, taking social developments and innovations into account (e.g., the Covid-19 pandemic) | Out (R2) | 39% (n = 25) |
| | 9 | Justification of the chosen method (Delphi procedure) to answer the research question | In (R2) | 83% (n = 57) |
| | 10 | Aim of the Delphi procedure (e.g., consensus, forecasting) | In (R1) | 89% (n = 81) |
| | | Information if the Delphi study is combined with another study (e.g., systematic review to develop the questionnaire, focus group with patients to discuss the Delphi results) | *No consensus* | 70% (n = 39) |

or additional information about project background need to be reported. In terms of an ethics vote, it is "typically not required to perform a Delphi in health sciences, since it does not involve human subjects" (free-text comment in the second Delphi round).

The experts did not agree to include any item from the section on *theory* in the reporting guideline (Table 4). In regard to the item about research paradigm, the free-text responses displayed opposing patterns of argument. Several of the respondents viewed Delphi studies as belonging to the quantitative paradigm ("A qualitative questionnaire is qualitative research, not Delphi"; commentary from the first Delphi round). For these experts, Delphi judgments have a universal and evidence-based character. Other respondents assigned Delphi studies to the qualitative paradigm ("A Delphi study has the aim to communicate and have a discussion, it is qualitative research"; commentary from the second Delphi round). This latter group emphasizes the relevance of open-ended questions in Delphi procedures, e.g., to gather context for specific judgments.

In the section covering *content*, justifying the selected method and stating the aim of a Delphi study are central elements of reporting (Table 4). What is not necessary, according to the respondents, is reporting within the context of current social developments. Disagreement remains about the items on making the relevance of a study clear. The argument against this is a pragmatic one, namely that a reporting guideline cannot cover all conceivable aspects.

## Topic: *Method*

The topic of *Method* was divided into seven sections: *body & integration of knowledge*, *Delphi variations*, *sample of experts*, *survey*, *Delphi rounds*, *feedback*, and *data analysis*. Consent was found for reporting on all three of the items asked about in the section on the *body & integration of knowledge* (Table 5), Accordingly, the identification of relevant expertise, the handling of missing knowledge, and an explanation of who is considered an expert in a particular Delphi study are considered important aspects when reporting a Delphi study.

In the section addressing *Delphi variations*, the experts agreed that it is important to identify and justify the Delphi variants and any modifications (Table 5).

In the section on the *sample of experts*, the selection criteria, how experts were found, and information about the recruitment process must be described (Table 5). How anonymity was handled was not viewed as relevant by the experts. The arguments in the free-text comments for disclosing respondents' identities included a better understanding of the judgments; the counterargument posed the question whether the relevant people would still participate in that case. Dissent remained concerning the relevance of reporting dropouts.

Eleven items were proposed in the section on *survey*, for which agreement on two items was reached (Table 5). The experts considered a general description of the questionnaire's development and the survey process to be relevant. What was found irrelevant or remained in dissent were, among other things, items regarding the pretest of the questionnaire and naming the software used.

In the sections about *Delphi rounds* and *feedback*, the experts agreed on reporting the number of rounds, identifying the objectives of each Delphi round, defining a termination criterion, and giving a detailed description of the feedback's design, including if group-specific analysis should be made available or, if applicable, how dissent was handled (Table 5).

In the section covering *data analysis*, it was agreed that the analytical methods applied to quantitative and qualitative data, the definition of consensus, and information regarding subgroup analysis or the weighting of the expert groups must be reported (Table 5). The percentage agreement for reporting the software used for analysis lies below the defined value for consensus.

**Table 5. Results for the topic of *Method*.**

| Section | No. | Checklist (= Items) | Consensus (Round) | Agreement % (n) |
|---|---|---|---|---|
| Body & Integration of knowledge | 11 | Identification and elucidation of relevant expertise, spheres of experience, and perspectives (e.g., theory, practice, affected groups, disciplines) | In (R1) | 78% (n = 69) |
| | 12 | Handling of knowledge, expertise and perspectives which are missing or have been deliberately not integrated | In (R1) | 75% (n = 66) |
| | 13 | Basic definition of expert[1] | In (R1) | 79% (n = 71) |
| Delphi variations | 14 | Identification of the type of Delphi procedure and potential modifications (e.g., classic Delphi, real-time Delphi, group Delphi) | In (R1) | 80% (n = 71) |
| | 15 | Justification of the Delphi variation and modifications, including during the Delphi process, if applicable | In (R1) | 79% (n = 70) |
| Sample of experts | 16 | Selection criteria for the experts (per round if there are different expert groups) | In (R2) | 94% (n = 65) |
| | 17 | Identification of the experts | In (R2) | 78% (n = 54) |
| | 18 | Information about recruiting and any subsequent recruiting of experts | In (R2) | 78% (n = 53) |
| | | *Information about how refusals and dropouts are handled (e.g., number of reminders, non-response analyses)* | *No consensus* | 73% (n = 41) |
| | | Anonymity of the experts | Out (R2) | 49% (n = 33) |
| Survey | 19 | Elucidation of the content development for the questionnaire[2] | In (R2) | 81% (n = 55) |
| | 20 | Description of the questionnaire (content and structure) | In (R3) | 86% (n = 48) |
| | | *Number of questions (open, closed, hybrid)* | *No consensus* | 66% (n = 37) |
| | | *Reference to additional integrated materials or information (e.g., info boxes illustrating the current knowledge about the theme focused on)* | *No consensus* | 52% (n = 28) |
| | | *Information about and justification of the types of scales used (e.g., nominal scales, rating or ranking scales)* | *No consensus* | 63% (n = 35) |
| | | Information about the graphic design of the questionnaire (e.g., use of figures) | Out (R2) | 28% (n = 19) |
| | | *Information about the validity of the items/scales (e.g., information on the piloting of the questionnaire or the evaluation of validity)* | *No consensus* | 56% (n = 31) |
| | | Information about the query regarding the experts' degree of certainty or competency | Out (R2) | 43% (n = 28) |
| | | Information about the pretest for the questionnaire | Out (R2) | 42% (n = 28) |
| | | Length of time to fill out the questionnaire per round | Out (R2) | 35% (n = 24) |
| | | Information about the software used for the survey (e.g., soscisurvey, e-delphi) | Out (R2) | 39% (n = 27) |
| Delphi rounds | 21 | Number of Delphi rounds | In (R1) | 88% (n = 80) |
| | 22 | Information about the aims of the individual Delphi rounds | In (R1) | 77% (n = 70) |
| | 23 | Disclosure and justification of the criterion for discontinuation | In (R1) | 83% (n = 74) |
| Feedback | 24 | Information about what data was reported back per round | In (R1) | 86% (n = 77) |
| | 25 | Information on how the results of the previous Delphi round were fed back to the experts surveyed (e.g., via frequencies, mean values, measures of dispersion, listing of comments) | In (R3) | 80% (n = 44) |
| | 26 | Information on whether feedback was differentiated by specific groups (e.g., by field of expertise, institutional affiliation) | In (R3) | 76% (n = 41) |
| | 27 | Information about how dissent and unclear results were handled | In (R1) | 86% (n = 78) |
| Data analysis | 28 | Disclosure of the quantitative and qualitative analytical strategy | In (R1) | 86% (n = 78) |
| | | *Information about the software used for analysis (e.g., SPSS, R, MAXQDA)* | *No consensus* | 54% (n = 30) |
| | 29 | Definition and measurement of consensus | In (R1) | 95% (n = 86) |
| | 30 | Information on group-specific analysis or weighting of experts (e.g., theory vs. practice, discipline-specific analysis) | In (R1) | 81% (n = 73) |

Previous reference in questionnaire: For us, "experts" are the participants; this can be people from academia, practice, or representatives of lived experience (e.g., patients, family members).

[2] *Note: We use the term "questionnaire" for the survey instrument regardless of whether quantitative or qualitative items are integrated or weighted.*

**Table 6. Results for the topic of *Results*.**

| Section | No. | Checklist (= Items) | Consensus (Round) | Agreement % (n) |
|---|---|---|---|---|
| Delphi process | 31 | Illustration of the Delphi process (e.g., in a flow chart) | In (R3) | 75% (n = 42) |
| | 32 | Information about special aspects during the Delphi process (e.g., deviations from the intended approach with justification) | In (R2) | 86% (n = 59) |
| | 33 | Number of experts per round (both invited and participating) | In (R1) | 88% (n = 80) |
| | | Information about the experts' sociodemographics per round | Out (R2) | 38% (n = 26) |
| | | *Information about expert competency (e.g., via professional experience, institutional affiliation, expertise in relevant fields/disciplines, conflict of interests)* | *No consensus* | 73% (n = 40) |
| Results | 34 | Presentation of the results for each Delphi round and the final results | In (R1) | 80% (n = 73) |

## Topic: *Results*

The topic involving *Results* contained the two sections on *Delphi process* and *results*. In the section on *Delphi process* there is consensus that the process, the number of experts per Delphi round, and any unexpected events during the Delphi process must all be reported (Table 6). Not included in the consensus are the reporting of sociodemographic characteristics and information about the experts' competency. Emerging from the free-text comments is the observation that it is difficult to define and measure competence.

In the section focused on *results* the experts argued for presenting the results of each round (Table 6).

## Topic: *Discussion and dissemination*

The topic of *Discussion and Dissemination* was subdivided into the two sections on *quality of findings* and *dissemination*. Belonging to the section on *quality of findings* is the reporting of a study's results, the validity and reliability of the findings, and possible limitations of a Delphi study (Table 7). With 74%, the agreement on the external validity of the results lies just under the cut-off value which requires 75% agreement.

No items from the section on *dissemination* will be included (Table 7).

## Discussion

The proposed reporting guideline for Delphi studies in the health and social sciences encompasses a total of 38 items that have been agreed upon by an international expert panel of Delphi practitioners. By including experts from different subject areas and with broad range of Delphi knowledge, we assume that the DELPHISTAR Reporting Guideline will be received very well by the scientific community. It is comparable in its scope to established guidelines, e.g.,

**Table 7. Results for the topic of *Discussion and dissemination*.**

| Section | No. | Checklist (= Items) | Consensus (Round) | Agreement % (n) |
|---|---|---|---|---|
| Quality of findings | 35 | Highlighting the findings from the Delphi study | In (R3) | 89% (n = 49) |
| | 36 | Validity of the results (e.g., transferability of the findings) | In (R1) | 78% (n = 69) |
| | 37 | Reliability of the results (e.g., how many people analyzed the qualitative responses) | In (R3) | 80% (n = 43) |
| | | *External validity of the findings* | *No consensus* | 74% (n = 39) |
| | 38 | Reflection on potential limitations (e.g., distortion, skewing, bias) | In (R1) | 89% (n = 81) |
| Dissemination | | *Availability of the dataset* | *No consensus* | 61% (n = 34) |
| | | Accessibility of the results for interested members of the public | Out (R2) | 49% (n = 34) |
| | | Information about further use of the results | Out (R2) | 42% (n = 29) |

CONSORT [64] (37 items) and PRISMA [51] (42 items). The requirement of 75% for a consensus resulted in the exclusion of several items that in some cases only very narrowly failed to meet this criterion; and in future discussions regarding the reporting guideline, it would be worth considering the possible inclusion of these items as "desirable" based on some type of grading system [65]. Ten items (e.g., external validity, information about expert competency) achieved a consensus ranging from more than 60% up to 74% in the third Delphi round. A consensus ranging between 50% and 59% was reached in the third round for five items (e.g., information about the software used for analysis, information about the validity of the items/ scales).

First and foremost, we expect an improvement in the reporting of Delphi studies. The potential for this is demonstrated by analyses of existing reporting guidelines, for instance, studies evaluating the Consolidated Standards of Reporting Trials (CONSORT) checklists show that the use of the reporting guideline is associated with an improved reporting of randomized controlled trials [66, 67]. We also expect to see a simplification or harmonization of the review process for Delphi studies and a raised awareness in Delphi practitioners about the quality of Delphi studies.

That said, the implementation of this recommended guideline is also contingent on whether journals require and check for the use of the guideline [67]. It is no less important for us, as the DEWISS Network, to promote DELPHISTAR to familiarize the target fields with it and to publish in the EQUATOR network. In terms of dissemination, we intend to create our own website, upload a short video via social media, and also inform the publishers of relevant journals and Delphi practitioners via email. Regarding this specific objective, the participating experts will be explicitly asked for their evaluation of the reporting guideline and their participation in the Delphi after the fact [52]. By doing this, we hope to gain information and insights concerning the quality of this Delphi study and future Delphi procedures.

Several items remained without agreement or did not meet the previously defined criterion for consent, the reason for which could possibly be traced back to the lack of methods research. This is seen in regard to three aspects:

1. The agreement to exclude items involving theory is a sign of absent discussions about the theoretical positioning of Delphi studies. Nonetheless, this would still be important because the definition of an epistemological aim is directly connected with the selection of quality criteria for Delphi studies [68]. Delphi studies that are more qualitative must be measured against criteria such as transparency or intersubjective comprehensibility; whereas quantitative Delphi studies have more to do with criteria such as scale quality and reliability of the results [23]. Admittedly, no established criteria yet exist to evaluate the quality of Delphi studies, even though initial proposals are available [52, 69].

2. The dissent around the items involving expert competency or scale validity could indicate that there is still too little methods research on this that investigates the potential influence of these aspects on judgement behavior and, thus, on the results [70].

3. Evaluations of Delphi studies could also provide new information. To date, such evaluations are carried out only in individual instances [71], but could yield important insights regarding the participants' motivations and judgment behaviors. This knowledge could also be relevant to further development of the Delphi reporting guideline.

We make the claim that DELPHISTAR can be used with different Delphi variants. Viewed from a quantitative perspective, it could be critically said that most of the participating experts consider their expertise to be in the classic Delphi, real-time Delphi, policy Delphi and the group Delphi. This was to be expected because, despite the increasing differentiations and

methodological modifications, these are the most frequently used Delphi variants [41, 72]. Argued from a qualitative standpoint, we assume based on our sampling method that the individually surveyed experts have a very high level of proficiency in the Delphi techniques covered by the questionnaire. Despite this, we are not able to determine with certainty that the items in the reporting guideline can be applied to all of the innovations and modifications to Delphi procedures. It is also for this reason that we plan to take a further step to test this reporting guideline on a defined random sample of publications in order to ensure feasibility.

## Strengths and limitations

The results of our Delphi survey must be viewed in the context of the expert panel and the survey time point in 2022. We assume that the use and applicability of DELPHISTAR must be subject to ongoing critical reflection. It is possible that items which were not included in the reporting guideline will be required by reviewers (e.g., "time period in which the Delphi study was conducted"). Furthermore, technical innovations, methodological developments and discussions regarding methods can affect Delphi studies thus changing the criteria for reporting them (e.g., "information about the software used for analysis"). This suggests that discussions about the participation of affected persons in Delphi studies conducted in clinical or nursing contexts will become increasingly more important, very possibly making methodological modifications to Delphi techniques necessary [73, 74]. Information regarding ethical approval would become much more important as a consequence.

In the Delphi study presented here, it was possible to achieve a typical response rate for international online Delphi studies, with approximately 10% [75]. Reasons why experts did not participate could involve language barriers or not receiving the emails. Using private email addresses for this would be conceivable, as several authors recommend [76]. It is possible that the regular reminder may have been effective in encouraging participation in all three Delphi rounds, in that, among other things, the actual completion time (average time for the experts participating up to that point) was included in the feedback.

The expert panel's geographic heterogeneity was successfully maintained. Nevertheless, biases in the panel could be present due to the predominance of experts with a background in the health sciences. Furthermore, only Delphi experts who published between 2016 and 2021 were included. It is possible that, as a consequence, specialists who also possess a high level of expertise and an impressive publication history in this field were excluded.

A relatively strict consensus criterion of 75% was selected for this Delphi study, which results in items being either kept or rejected. Considerations could have been made to divide the results into different categories, for example, into three categories with a) items of highly consensual and necessary inclusion (e.g., 75% and above), b) items of desirable and generally necessary inclusion (e.g., between 60% and 75%), and c) possible items of inclusion depending on the study and study objectives (less than 60%). Following this strategy may very well have produced a differentiated yet more complex reporting guideline.

## Supporting information

**S1 File. Delphi studies in social and health sciences–recommendations for an interdisciplinary standardized reporting (DELPHISTAR).**
(DOCX)

**S2 File. DELPHISTAR–questionnaires and datasets of the Delphi rounds.**
(ZIP)

## Acknowledgments

We extend our thanks to all of the experts who participated in the Delphi study and contributed to the development of a reporting guideline for Delphi studies. The following lists all of the experts who participated in the survey rounds and gave their consent to be named.

We are grateful for the time, commitment, and expertise of the following members of the Delphi Expert Panel. All experts who participated at all survey rounds and wished to be acknowledged are named below.

Named with permission (not all Delphi participants wished to be acknowledged):

Alam, M., Department of Dermatology, Northwestern University, USA

Backman, C., The University of British Columbia, Canada

Banno, M., Department of Psychiatry, Seichiryo Hospital, Japan

Bartoszko, J., Department of Anesthesia and Pain Management, Toronto General Hospital-University Health Network, Canada

Bloomfield, F., Liggins Institute, University of Auckland, New Zealand

Bober, M. B., Division of Orthogenetics, Nemours A.I. duPont Hospital for Children, USA

Chalkoo, M., Government Medical College Srinagar Kashmir, India

Chan, T. M., McMaster University, Canada

Chen, Y., Evidence-Based Medicine Center, School of Basic Medical Sciences, Lanzhou University, China

Coscia, C., Department of Architecture and Design (DAD), Politecnico di Torino, Italy

de Luca, K., Discipline of Chiropractic, School of Health, Medical and Applied Sciences, CQUniversity, Australia

Demartines, N., University Hospital CHUV of Lausanne, Switzerland

Farzandipour, M., Kashan University of Medical Sciences, Iran

Goldstein, D., University of New South Wales, Australia

Goodman, C., Centre for Research in Public Health and Community Care (CRIPACC), University of Hertfordshire, United Kingdom

Grant, S., HEDCO Institute for Evidence-Based Educational Practice, College of Education, University of Oregon, USA

Hübner, M., Centre hospitalier universitaire vaudois, Switzerland

Harris, T., Polycystic Kidney Disease Charity, United Kingdom

Howell, M., University of Sydney, Australia

Huber, A. M., IWK Health and Dalhousie University, Canada

Jansen, T. L., VieCuri Medisch Centrum, Netherlands

Johnson, N., Consultant hand surgeon, Pulvertaft Hand Centre, United Kingdom

Julious, S. A., School of Health and Related Research, University of Sheffield, United Kingdom

Kenny, G. P., Human and Environmental Physiology Research Unit, University of Ottawa, Canada

Konge, L., Copenhagen Academy for Medical Education and Simulation (CAMES), University of Copenhagen, Denmark

Kopkow, C., Center for Evidence-Based Healthcare, University Hospital and Medical Faculty Carl Gustav Carus, TU Dresden, Germany

LaPrade, R. F., Twin Cities Orthopedics, USA

Lim, M., Evelina London Children's Hospital; King's Health Partners Academic Health Science Centre; Department Women and Children's Health, School of Life Course Sciences (SoLCS), Faculty of Life Sciences and Medicine, Kings College London, United Kingdom

Loudovici-Krug, D., ÄMM Research Consultation Center, Institute of Physical and Rehabilitative Medicine, University Hospital Jena, Germany

Ma, Y., Department of Nursing, Chinese PLA General Hospital, China

MacLennan, S., Academic Urology Unit, Institute of Applied Health Sciences, University of Aberdeen, United Kingdom

Mokkink, L. B., Department Epidemiology and Data Science, Amsterdam UMC, Vrije Universiteit Amsterdam, Netherlands

Montero, M., Department of Environment, Centro de Investigaciones Energéticas, Medioambientales y Tecnológicas (CIEMAT), Spain

Myles, P. S., Monash University, Australia

Nayahangan, L. J., Center for Human Resources and Education, Copenhagen Academy for Medical Education and Simulation (CAMES), Denmark

Pace, N. L., University of Utah, USA

Page, M. J., Monash University, Australia

Parente, F., Psychology Department, Towson University, USA

Payne, K., The University of Manchester, United Kingdom

Petrovic, M., Department of Internal Medicine and Paediatrics, Ghent University, Belgium

Sprung, C. L., Hadassah Medical Organization and Faculty of Medicine, Hebrew University of Jerusalem, Israel

Raveendran, K., Fatimah Hospital, Malaysia

Roller-Wirnsberger, R., Department of Internal Medicine Graz, Medical University of Graz, Austria

Sconfienza, L. M., University of Milano, Italy

Spinelli, A., Department of Biomedical Sciences, Humanitas University, Italy

van der Heijde, D., Leiden University Medical Center, Netherlands

Vohra, S., University of Alberta, Canada

Weissman, J. S., Center for Surgery and Public Health, Brigham and Women's Hospital, Harvard Medical School, USA

Westby, M., Centre for Aging SMART Vancouver, Canada

Wu, Y., Peking University School of Public Health and Clinical Research Institute, China

Yadlapati, R., Division of Gastroenterology, University of California San Diego, USA

Yarris, L. M., Oregon Health & Science University, USA

Zhang, X., Hong Kong Baptist University, China

## Author Contributions

**Conceptualization:** Marlen Niederberger.

**Data curation:** Marlen Niederberger, Julia Schifano.

**Formal analysis:** Marlen Niederberger, Julia Schifano, Stefanie Deckert, Julian Hirt, Angelika Homberg, Stefan Köberich, Rainer Kuhn, Alexander Rommel, Marco Sonnberger.

**Funding acquisition:** Marlen Niederberger.

**Investigation:** Marlen Niederberger.

**Methodology:** Marlen Niederberger, Julia Schifano.

**Project administration:** Marlen Niederberger.

**Resources:** Marlen Niederberger.

**Software:** Marlen Niederberger, Julia Schifano.

**Supervision:** Marlen Niederberger.

**Validation:** Marlen Niederberger.

**Visualization:** Marlen Niederberger, Julia Schifano.

**Writing – original draft:** Marlen Niederberger, Julia Schifano, Stefanie Deckert, Julian Hirt, Angelika Homberg, Stefan Köberich, Rainer Kuhn, Alexander Rommel, Marco Sonnberger.

**Writing – review & editing:** Marlen Niederberger, Julia Schifano.

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
