## [Decision Letter · Decision Letter 0]

23 Jan 2024

PONE-D-23-30916Delphi studies in social and health sciences – recommendations for an interdisciplinary standardized reporting (DELPHISTAR). Results of a Delphi study.PLOS ONE

Dear Dr. Niederberger,

Thank you for submitting your manuscript to PLOS ONE. After careful consideration, we feel that it has merit but does not fully meet PLOS ONE’s publication criteria as it currently stands. Therefore, we invite you to submit a revised version of the manuscript that addresses the points raised during the review process. Although the manuscript presents research with interest to the academic and practitioners' community, as well as is of interest to PLOS ONE readers, some changes are required to improve its clarity, accuracy and depth of analysis.

We look forward to receiving your revised manuscript.

Kind regards,

Monica Duarte Correia de Oliveira

Academic Editor

PLOS ONE

Journal Requirements:

3. Please provide additional details regarding participant consent. In the ethics statement in the Methods and online submission information, please ensure that you have specified (1) whether consent was informed and (2) what type you obtained (for instance, written or verbal, and if verbal, how it was documented and witnessed).

4. Thank you for stating the following financial disclosure: "all authors Deutsche Forschungsgemeinschaft (DFG) - Projektnummer 429572724 Network promotion". 

5. In the online submission form, you indicated that "The data underlying the results presented in the study are available from Prof. dr. Marlen Niederberger (marlen.niederberger@ph-gmuend.de)".

Additional Editor Comments:

The study is on a relevant topic for the PLOS One audience. Nevertheless, in my opinion the manuscript needs to be revised before it can be considered for publication by the journal. Namely:

MAJOR COMMENTS

The study has developed an e-Delphi, which was also described as a classic Delphi. The authors should describe better what is an e-Delphi, and rethink whether the study is a classic Delphi given the features that follow.

There is an adoption of features in the Delphi study that raise issues – usually the items do not change in sequential rounds, but the authors describe “The initial questionnaire was revised based on the results of the first Delphi round, meaning that consented items were removed and the remaining items were reworded as necessary”. However rewording of items in sequential rounds raises issues for interpretation, comparability and analyses which are not reflected upon in the manuscript. Literature should frame why the procedure is acceptable, and I am not sure whether this is a classical Delphi, or its links to an enchained Delphi. It would be important to understand whether participants were informed about the selected majority rules prior to the Delphi.

The authors need to explain how Table 1 was generated.

The authors need to explain the rationale and roots for using the selected importance scale.

The discussion on whether the proposed guideline is useful for distinct types of Delphi process types/variants needs to be extended and deepened (classical Delphi, policy Delphi, etc).

I do not understand the appropriateness of using the sentence “None of the experts faced any repercussions for deciding not to participate or for dropping out”.

As the manuscript claims that a key output is a complete guideline list, I wonder whether it be presented or summarized within the manuscript.

The manuscript does not discuss future research (following the work done), as well as should go deeper in the study limitations.

The authors should provide more detailed information on the search protocol used to get the two Zotero databases.

MINOR COMMENTS

I missed an explicit statement of the objectives before the methods section.

The authors should comment on what could be done to avoid “some of the emails were blocked”.

Reviewers' comments:

Reviewer's Responses to Questions

**Comments to the Author**

1. Is the manuscript technically sound, and do the data support the conclusions?

Reviewer #1: Yes

Reviewer #2: Yes

2. Has the statistical analysis been performed appropriately and rigorously? 

Reviewer #1: Yes

Reviewer #2: Yes

3. Have the authors made all data underlying the findings in their manuscript fully available?

Reviewer #1: Yes

Reviewer #2: Yes

4. Is the manuscript presented in an intelligible fashion and written in standard English?

Reviewer #1: Yes

Reviewer #2: Yes

5. Review Comments to the Author

Reviewer #1: PLOS One

Paper title: Delphi studies in social and health sciences – recommendations for an interdisciplinary standardised reporting (DELPHISTAR). Results of a Delphi study.

Ref: PONE-D-23-30916

Thank you for allowing me the opportunity to review this paper it was a delight to read. The study adds to the evidence based relating to the reporting of Delphi Studies. Whilst such guidelines do exist, this study builds upon this evidence and recent application examples to formulate a 3-round modified Delphi. The authors should be commended for this in-depth approach adopted. See below some minor reflections/ suggestions:

As an observation, the rationale underpinning the use of guidelines, could be brought stronger to the fore for example, to address methodological misunderstandings, enhance the rigour and quality. Moreover, this study proposes addressing the need for nonspecific areas but generically across health and social are sciences needs to be brought to the fore in the background, as well as consideration for why this wide area needs this evidence (i.e., most utilised within this field?).

The low uptake of the Delphi within this study from the total sample targeted also needs to be addressed and may be related to the inclusion criteria required to read and speak English, hence limiting participation.

There is a sense from the dicussion that this should be applied but dependent upon journals acceptability. However, would further expert panel consultation not also be beneficial, prior to implementation? Moreover, how authors apply this may also need to be developed. We assume the existing frameworks are appropriate but some blue sky thinking of how to move these forwards are rarely discussed.

Reviewer #2: The article presented is an interesting and necessary contribution to the study and improvement of the Delphi methodology. It is the result of a systematic and well-oriented research process, which has concluded with a Delphi study to propose the most important items, agreed with the scientific community that usually uses this technique, which should be included in the reporting of a Delphi study. This is the next logical and necessary step after the previous study carried out by the Niederberger and Spranger team (2020), which had conducted a systematic review of the methodological work published to date on the Delphi method and on the elements that the various authors recommended to include in the reporting of results.

As could not be otherwise, in this study the method has been used with rigor, the group of experts consulted has been large and justified, the number of responses obtained is sufficient, the results are valuable and the reporting is very well done.

Nevertheless, we would like to make some comments or suggestions, in the hope that the authors will assess their applicability to the improvement of this work.

- Selection of experts. The expert selection process is perfectly justified and defined in the report. However, the decision criteria selected may, in our opinion, have excluded potential quality contributions. In other words, all participants are experts, but perhaps not all genuine experts were able to participate. The criterion of limiting the search to authors who have published between 2016 and 2021 may have left out relevant authors. In the list of published experts I miss the main methodologists of this technique, alive, according to the number of citations that their works have accumulated: Rowe, Wright, Okoli, Pawlowski, Adler, Ziglio, Skulmoski, Krann, Landeta, Von der Gracht, Gordon, Pease, Tapio, Turoff, Hasson... They may have been invited and declined to participate, or they did not want to publicize their names, but the reality is that none of the most cited appears. Moreover, the profile of the experts consulted corresponds mainly to Health Sciences, and very little to Social Sciences. It is not a question of repeating the study, but of indicating more clearly this deficiency in the final limitations.

- In order to accept the inclusion of an item in the final list of items to be included in a report, the criterion was that more than 75% of the experts who responded rated its importance with 6 or 7 points (out of 7). This is a criterion, but I believe it is too restrictive and leads to the loss of valuable information. It is true that in the tables the % of consensus on each item is maintained, but the presentation of results simplifies the analysis with a Yes or No item.

In my opinion, all the items included (65) have their importance and their reason to be in a final report, even some more could have been. Therefore, the results should be, at least, classified in three categories:

a. Items of highly consensual, necessary and recommended inclusion (e.g., above 75% consensus).

b. Items of desirable and generally necessary inclusion (e.g., between 50% and 75%).

c. Possible inclusion items, depending on the study and study objectives (less than 50%).

Or something similar.

- The contribution of the work is valuable and necessary, but at the end of the reading a slight disappointment remains, because a study carried out with such rigor and with the participation of so many experts is limited to providing a list of items to be included in a reporting hierarchy according to the degree of consensus they have reached. We receive no information on the reasoning that these people have used to support their position in favor, or not, of considering each item as very important. I would ask the authors to provide in an auxiliary document the main arguments for and against the inclusion of each item in the final list, gathered from the qualitative contributions of the experts.

- There are two final items that have obtained a high consensus Reliability of the results (80%) and External validity of the findings (74%) that could have been worked on more in this study, providing indicators of the quality of the work based on judgments external to the authors. For example, including a survey of the participating experts in which they are asked about the rigor with which the study was conducted, their confidence in the results, their satisfaction with the participation... A recent publication on this subject is Landeta and Lertxundi (2023). Quality indicators for Delphi studies. Futures & Foresight Science, e172.

In summary, it is an interesting and necessary article, which could be improved by considering the comments made on the limitations of its expert participants, the classification of the items analyzed, the additional qualitative information that could be provided and the inclusion of some indicator of the quality of the study external to the authors who carried it out.

6. PLOS authors have the option to publish the peer review history of their article (what does this mean?). If published, this will include your full peer review and any attached files.

Reviewer #1: **Yes: **Dr Felicity Hasson

Reviewer #2: **Yes: **Jon Landeta

---

## [Author Response · Author response to Decision Letter 0]

20 Mar 2024

Dear editor,

Dear Dr. Felicity Hasson,

Dear Professor Jon Landeta,

Thank you for your valuable insights and comments on our manuscript, "Delphi studies in social and health sciences – recommendations for an interdisciplinary standardized reporting (DELPHISTAR). Results of a Delphi study." We have given your constructive feedback careful consideration and used it to improve the quality and clarity of our paper. The comments regarding the detailed description of the methodological approach have improved the transparency and understandability of our approach. Above all, reflecting on the study's limitations and the discussion section, we found your remarks to be a valuable way to expand on how to interpret the study results.

We very much appreciate the time and effort that you have spent evaluating our manuscript, and we are happy to send you now the revised version.

With kind regards,

Prof. Dr. Marlen Niederberger on behalf of the authors

---

## [Decision Letter · Decision Letter 1]

26 Apr 2024

PONE-D-23-30916R1Delphi studies in social and health sciences – recommendations for an interdisciplinary standardized reporting (DELPHISTAR). Results of a Delphi study.PLOS ONE

Dear Dr. Niederberger,

Thank you for submitting your manuscript to PLOS ONE. After careful consideration, we feel that it has merit but does not fully meet PLOS ONE’s publication criteria as it currently stands. Therefore, we invite you to submit a revised version of the manuscript that addresses the points raised during the review process.

The authors have improved the manuscript, considering all suggestions and comments, and the manuscript now provides a sound reporting guideline (DELPHISTAR) that will be helpful for those developing Delphi studies. Before acceptance for publication, the authors should make the minor revision suggested by the referee. ==============================

We look forward to receiving your revised manuscript.

Kind regards,

Monica Duarte Correia de Oliveira

Academic Editor

PLOS ONE

Journal Requirements:

Reviewers' comments:

Reviewer's Responses to Questions

**Comments to the Author**

1. If the authors have adequately addressed your comments raised in a previous round of review and you feel that this manuscript is now acceptable for publication, you may indicate that here to bypass the “Comments to the Author” section, enter your conflict of interest statement in the “Confidential to Editor” section, and submit your "Accept" recommendation.

Reviewer #2: All comments have been addressed

2. Is the manuscript technically sound, and do the data support the conclusions?

Reviewer #2: Yes

3. Has the statistical analysis been performed appropriately and rigorously? 

Reviewer #2: Yes

4. Have the authors made all data underlying the findings in their manuscript fully available?

Reviewer #2: Yes

5. Is the manuscript presented in an intelligible fashion and written in standard English?

Reviewer #2: Yes

6. Review Comments to the Author

Reviewer #2: The authors have successfully incorporated the recommendations suggested by the reviewers. The work is now clearer, more open, more self-critical and more transparent.

The decision to make the supporting information of the study available to readers is appreciated.

In sum, it is a rigorous work that makes a necessary and valuable methodological contribution to the development and scientific consolidation of the Delphi method.

A minor comment:

In the revised text the first necessary feature of a Delphi study "Experts are surveyed while typically preserving their anonymity" (line 24 of the first draft) has been deleted. I assume this is a mistake. In addition, both line 24 and line 135 of the corrected manuscript still refer to the five typical characteristics of Delphi studies, despite the fact that only four are included in the new version.

As far as I am concerned, once this error has been corrected, the article is perfectly publishable.

Thank you very much.

7. PLOS authors have the option to publish the peer review history of their article (what does this mean?). If published, this will include your full peer review and any attached files.

Reviewer #2: No

---

## [Author Response · Author response to Decision Letter 1]

13 May 2024

Dear editor,

Dear reviewer,

Thank you for your prompt response and positive evaluation of our paper "Delphi studies in social and health sciences – recommendations for an interdisciplinary standardized reporting (DELPHISTAR). Results of a Delphi study". We have re-added the missing item to the list of Delphi characteristics and are grateful to you for drawing our attention to it. Furthermore, we have minimally changed the phrasing in several sentences. All of the changes are highlighted in color.

Overall, we want to thank you for the helpful information and respect shown during the entire review process. We are happy to have the paper published soon.

With kind regards,

Prof. Dr. Marlen Niederberger on behalf of the authors

---

## [Editor Report · Decision Letter 2]

16 May 2024

Delphi studies in social and health sciences – recommendations for an interdisciplinary standardized reporting (DELPHISTAR). Results of a Delphi study.

PONE-D-23-30916R2

Dear Dr. Niederberger,

We’re pleased to inform you that your manuscript has been judged scientifically suitable for publication and will be formally accepted for publication once it meets all outstanding technical requirements.

Kind regards,

Monica Duarte Oliveira

Academic Editor

PLOS ONE
---

## [Editor Report · Acceptance letter]

16 Jul 2024

PONE-D-23-30916R2 

PLOS ONE

Dear Dr. Niederberger, 

I'm pleased to inform you that your manuscript has been deemed suitable for publication in PLOS ONE. Congratulations! Your manuscript is now being handed over to our production team.

Kind regards, 

on behalf of

Professor Monica Duarte Correia de Oliveira 

Academic Editor

PLOS ONE